# Advancing Patient Safety: The Future of Artificial Intelligence in Mitigating Healthcare-Associated Infections: A Systematic Review

**DOI:** 10.3390/healthcare12191996

**Published:** 2024-10-06

**Authors:** Davide Radaelli, Stefano Di Maria, Zlatko Jakovski, Djordje Alempijevic, Ibrahim Al-Habash, Monica Concato, Matteo Bolcato, Stefano D’Errico

**Affiliations:** 1Department of Medical Surgical and Health Sciences, University of Trieste, 34127 Trieste, Italy; davide_radaelli@hotmail.it (D.R.); stefano.dimaria@studenti.units.it (S.D.M.); monica.concato@studenti.units.it (M.C.); 2Institute of Forensic Medicine, Criminalistic and Medical Deontology, University Ss. Cyril and Methodius, 1000 Skopje, North Macedonia; drjakovski@gmail.com; 3Institute of Forensic Medicine ‘Milovan Milovanovic’, School of Medicine, University of Belgrade, 11000 Belgrade, Serbia; djordje.alempijevic@med.bg.ac.rs; 4Forensic Medicine Department, Mutah University, Karak 61710, Jordan; ibrhbsh_forensic@yahoo.com; 5Department of Medicine, Saint Camillus International University of Health and Medical Sciences, 00131 Rome, Italy

**Keywords:** artificial intelligence, machine learning, healthcare-associated infections, infection prevention

## Abstract

Background: Healthcare-associated infections are infections that patients acquire during hospitalization or while receiving healthcare in other facilities. They represent the most frequent negative outcome in healthcare, can be entirely prevented, and pose a burden in terms of financial and human costs. With the development of new AI and ML algorithms, hospitals could develop new and automated surveillance and prevention models for HAIs, leading to improved patient safety. The aim of this review is to systematically retrieve, collect, and summarize all available information on the application and impact of AI in HAI surveillance and/or prevention. Methods: We conducted a systematic review of the literature using PubMed and Scopus to find articles related to the implementation of artificial intelligence in the surveillance and/or prevention of HAIs. Results: We identified a total of 218 articles, of which only 35 were included in the review. Most studies were conducted in the US (n = 10, 28.6%) and China (n = 5; 14.3%) and were published between 2021 and 2023 (26 articles, 74.3%) with an increasing trend over time. Most focused on the development of ML algorithms for the identification/prevention of surgical site infections (n = 18; 51%), followed by HAIs in general (n = 9; 26%), hospital-acquired urinary tract infections (n = 5; 9%), and healthcare-associated pneumonia (n = 3; 9%). Only one study focused on the proper use of personal protective equipment (PPE) and included healthcare workers as the study population. Overall, the trend indicates that several AI/ML models can effectively assist clinicians in everyday decisions, by identifying HAIs early or preventing them through personalized risk factors with good performance. However, only a few studies have reported an actual implementation of these models, which proved highly successful. In one case, manual workload was reduced by nearly 85%, while another study observed a decrease in the local hospital’s HAI incidence from 1.31% to 0.58%. Conclusions: AI has significant potential to improve the prevention, diagnosis, and management of healthcare-associated infections, offering benefits such as increased accuracy, reduced workloads, and cost savings. Although some AI applications have already been tested and validated, adoption in healthcare is hindered by barriers such as high implementation costs, technological limitations, and resistance from healthcare workers. Overcoming these challenges could allow AI to be more widely and cost-effectively integrated, ultimately improving patient care and infection management.

## 1. Introduction

Previously known as “nosocomial infections”, healthcare-associated infections (HAIs) are defined by the World Health Organization (WHO) as infections that patients acquire during medical or surgical treatment in hospitals or while receiving healthcare in other facilities (e.g., long-term care, family medicine clinics, home care, and ambulatory care), which were not evident or incubating at the time of admission [1]. These infections are among the most common adverse outcomes in healthcare, despite being largely preventable, and are often used as a measure of healthcare quality [2]. HAIs endanger patient safety, causing unnecessary deaths, prolonged hospitalizations, long-term disabilities, and increased antibiotic resistance among bacteria. This, in turn, leads to significant extra costs and financial strain on both the healthcare system and the patients’ families [2]. Assessing the impact of HAIs is challenging due to a lack of high-quality data, but it is believed that 2.6 million HAI cases occur annually in the European Union and European Economic Area, resulting in roughly 2.5 million disability-adjusted life years (DALYs) [3]. In the United States, the US Center for Disease Control and Prevention (CDC) reports that nearly 1.7 million hospitalized patients contract HAIs each year while receiving treatment for other conditions, with over 98,000 of these cases—approximately 1 in 17—resulting in death [4]. These data place HAIs among the top ten leading causes of mortality in the country [4]. On a global scale, 1 in every 10 patients is affected by HAIs [1]. Out of every 100 patients, 7 in advanced countries and 10 in emerging countries may acquire an HAI, especially those admitted to intensive care units [1]. The economic impact of HAIs is considerable, with costs estimated at 28-to-45 billion USD per year in the United States [4], and around 7 billion EUR annually in Europe, though the latter figure may be significantly underestimated [5].

Given the significant economic and clinical burden that these infections impose on society, the goal should be to prevent avoidable infections by employing effective surveillance strategies for infection prevention and control. A 2014 study revealed that infection preventionists—professionals dedicated to ensuring healthcare workers and patients adhere to infection prevention protocols—spent on average about half of their work hours on surveillance-related activities [6]. There are still multiple limitations in HAI surveillance, as its accuracy heavily depends on the training of infection control practitioners and is prone to subjective interpretation and surveillance bias.

A report by the European Centre for Disease Prevention and Control (ECDC) on HAIs and antimicrobial use in European acute care hospitals during 2022–2023 highlights a varying degree of automation in HAI surveillance across Europe. The percentage of hospitals with any level of automated HAI surveillance ranged from 0 to 20% in the Balkan region, 20 to 40% in countries like Germany and Slovakia, and over 80% in Finland and up to 100% in Iceland [7]. Currently, the most frequently targeted HAIs for automated surveillance are Clostridium difficile infections and bloodstream infections, with the highest automation rates at 36.3% and 36.7%, respectively, while hospital-acquired pneumonia had the lowest rate at 30.4%. The feasibility of implementing automated surveillance was also assessed by determining whether key variables or data sources for automation were available in digital format, and if so whether they were structured and well defined. Key administrative data (e.g., admission and discharge dates) were most likely to be available in structured digital formats, while data on the use of invasive devices (e.g., mechanical ventilation) had the lowest availability, with only 57.6% of hospitals reporting such data in digital form [7].

The rise of artificial intelligence (AI) and all its subsets, especially machine learning (ML), has the potential to improve and even revolutionize healthcare, thanks to the medical and technological advancements in the last decades and the availability of data coming from the increasingly widespread electronic health records (HERs). Some of the already tested applications of AI in healthcare that have proven useful include the detection of clinical conditions in medical imaging and diagnostic services (providing a reduction in diagnostic errors). In neurology and neurosurgery, for example, AI can improve risk management and surgical decision making by predicting postoperative complications such as infections, bleeding, and neurological deficits, thereby enhancing patient safety [8]. Additionally, numerous AI models can forecast stroke risk and reduce stroke-related deaths and disability burdens by analyzing various health feature analyses into an ensemble machine (Neuro-Health Guardian) [9].

Other benefits include improving patient compliance and engagement, reducing the time professionals spend on administrative duties, aiding in the development of new drugs, and, during the COVID-19 outbreak, enabling early diagnosis, patient monitoring, and management through virtual patient care [10]. Artificial intelligence has the potential to enhance productivity and improve the quality of care in two main ways: *information synthesis*, as the amount and complexity of data (e.g., patient data from electronic health records, gene sequencing, or medical literature) is overwhelming for a human operator to handle alone; and *enhancement of human performance*, by helping clinicians track and analyze all available information [11].

The previously mentioned substantial economic and clinical burden of HAIs underscores the urgent need for effective infection prevention and control strategies. Monitoring HAIs is essential for developing, implementing, and sustaining effective infection prevention and control programs. It is precisely in this context that AI offers a promising tool.

This review aims to systematically retrieve, collect, and summarize all available information on the application and impact of AI in HAI surveillance over the past decade and assess where and how AI and its various subsets (especially ML) have been trained and implemented and evaluate their performance.

Advancements in AI are not only improving diagnostic and management capabilities in healthcare but also showing promise in addressing ongoing challenges, such as HAIs.

## 2. Materials and Methods

A systematic literature review was conducted according to the Appendix A PRISMA 2020 guidelines. PubMed and Scopus were used to find articles related to the implementation of artificial intelligence in the surveillance and/or prevention of HAIs. The aim was to synthesize the main data published in the literature over the past ten years.

In the identification phase, the following combinations of keywords found within the titles and/or abstracts of the articles were used for PubMed: (ai OR artificial intelligence OR machine learning) AND healthcare-associated infections; (ai OR artificial intelligence) AND healthcare-associated infections; (ai OR artificial intelligence OR machine learning) AND nosocomial infections; (ai OR artificial intelligence) AND nosocomial infections; (ai OR artificial intelligence) AND surgical site infection; (ai OR artificial intelligence OR machine learning) AND hospital-acquired urinary tract infections; and (ai OR artificial intelligence) AND hospital-acquired pneumonia.

For Scopus, the following search strings were used: TITLE-ABS (ai) OR TITLE-ABS (artificial AND intelligence) AND TITLE-ABS (healthcare AND associated AND infections) AND PUBYEAR > 2013 AND PUBYEAR < 2025 AND (LIMIT-TO (SUBJAREA, “MEDI”)) AND (LIMIT-TO (DOCTYPE, “ar”)) AND (LIMIT-TO (LANGUAGE, “English”)); TITLE-ABS (ai) OR TITLE-ABS (artificial AND intelligence) AND TITLE-ABS (surgical AND site AND infections) AND PUBYEAR > 2013 AND PUBYEAR < 2025 AND (LIMIT-TO (SUBJAREA, “MEDI”)) AND (LIMIT-TO (DOCTYPE, “ar”)) AND (LIMIT-TO (LANGUAGE, “English”)); TITLE-ABS (ai) OR TITLE-ABS (artificial AND intelligence) AND TITLE-ABS (hospital AND acquired AND urinary AND tract AND infections) AND PUBYEAR > 2013 AND PUBYEAR < 2025 AND (LIMIT-TO (SUBJAREA, “MEDI”)) AND (LIMIT-TO (DOCTYPE, “ar”)) AND (LIMIT-TO (LANGUAGE, “English”)); and TITLE-ABS (ai) OR TITLE-ABS (artificial AND intelligence) AND TITLE-ABS (hospital AND acquired AND pneumonia) AND PUBYEAR > 2013 AND PUBYEAR < 2025 AND (LIMIT-TO (SUBJAREA, “MEDI”)) AND (LIMIT-TO (DOCTYPE, “ar”)) AND (LIMIT-TO (LANGUAGE, “English”)).

### 2.1. Criteria for Including/Excluding Studies

Articles in English, published in the last ten years (February 2014–February 2024), were included if the full text was available and they (1) presented the implementation of artificial intelligence or machine learning in the surveillance and/or prevention of HAIs; (2) reported original data.

### 2.2. Characteristics of Eligible Studies

A total of 218 articles were identified. Before the screening phase, duplicates were removed (n = 48). Subsequently, 170 articles were subjected to the screening phase: 116 articles were excluded as they were deemed irrelevant based on the title and/or abstract review, 15 were reviews, and 4 articles did not have the full-text version available. S.D.M. and M.C. were tasked with the assessment of the eligibility of the studies.

### 2.3. Quality Assessment and Risk of Bias

S.D.M. and M.C. independently assessed the abstracts and subsequently the full texts of the articles included in the review. The primary risk of bias was associated with the selection of the keywords for the search. No significant disagreements arose regarding the inclusion or exclusion of the articles and all studies included were considered of high quality and complete.

## 3. Results

A total of 35 articles were included in the systematic review. The flowchart in Figure 1 summarizes the flow of information through the different phases of the systematic review.

### 3.1. Data Extraction

The collected data are summarized in Table 1. The essential information includes the study setting and population, with its objectives and conclusions (such as performance, e.g., the area under the receiver operator curve, area under the curve, accuracy, sensitivity, specificity, and other findings), the specific infection targeted for prevention or surveillance, as well as the methodology used for AI training (including sample size, study duration, analyzed variables, and dataset source).

The articles included in the review were conducted across various countries. Most studies were from the USA [12,13,14,15,16,17,18,19,20,21] (n = 10, 28.6%), followed by China [22,23,24,25,26,27] with six studies (14.3%), and Denmark [28,29,30,31] with four studies (11.4%). Italy [32,33,34] and Canada [35,36,37] each contributed three studies. Brazil [38,39] and Japan [40,41] each provided two, and South Korea [42], Pakistan [43], Taiwan [44], the United Kingdom [45], and Spain [46] each had one study.

The included studies were published between 2017 and 2024, with the majority of them published between 2021 and 2023 (26 articles, 74.3%).

More than half of the articles (n = 22, 62.9%) did not specify the department where the study was conducted and gathered data from patients across the entire hospital. The departments most frequently involved in the studies were surgical units (n = 6, 17.1%), especially general surgery, orthopedics, and gynecology, as well as the intensive care unit (n = 4, 11.4%). Only two studies were conducted in pediatric departments, and one study focused on a psychiatric hospital. In most of the studies reviewed (n = 14, 40%), the data came from patients who had undergone some type of surgery (especially colon surgery, n = 4, 11.4%). In nine articles, however, the study population included all patients admitted to a particular hospital during a specific period, regardless of their conditions or departments. Interestingly, in only one instance, the study focused on the hospital staff to assess how proper handwashing and the procedures for putting on and taking off personal protective equipment affected the rate of hospital-acquired infections [27].

Most of the data used to train the AI and its different models came from internal sources within the hospital, such as electronic and paper health records, clinical or surgical notes, and lab results of the patient subjects of the study (n = 28, 80%), in some cases even using an AI model (natural language processing—NLP) to retrieve said data from clinical notes [12]. In the remaining cases, the data came from a combination of internal records and publicly available databases, or exclusively from the latter. These included resources like the MIMIC dataset, the eICU Collaborative Research Database, and the American College of Surgeons National Quality Improvement Program database [40].

### 3.2. Types of HAIs

To improve clarity and facilitate understanding, the studies and their outcomes have been categorized by HAI type: surgical site infections (SSIs), healthcare-associated pneumonia (HCAP), hospital-acquired urinary tract infections (HA-UTIs), and miscellaneous HAIs. Figure 2 displays the number of studies conducted for each infection type.

#### 3.2.1. Surgical Site Infections—SSIs

Surgical site infections are among the most common types of hospital-acquired infections, being the most frequent infection afflicting surgical patients, accounting for nearly 20% of all HAIs in European hospitals. SSIs are the costliest type of HAI, increasing the financial burden of surgery by leading to longer hospital stays, additional diagnostic tests, treatments, and frequently the need for further surgical procedures [47,48]. It is not surprising, then, that our review of the literature found that most studies involving AI, especially those using ML, were focused on developing predictive models for the early detection and prevention of SSIs (18 studies, 51% of the total articles reviewed).

SSIs following colorectal surgery received particular attention [9,18,38,39]. Sohn et al. [12] developed an automated Bayesian network system that uses risk factors from the American College of Surgeons National Surgical Quality Improvement Program (ACS NSQIP) and data extracted from clinical notes of surgical procedures using an NLP model. The system was able to identify SSIs with a Receiver Operating Characteristic (ROC) of 0.827, which increased to 0.892 when surgeons helped the AI to identify clinically meaningful SSIs.

Another team [30] developed a natural language processing (NLP) model able to read electronic health record chart notes and predict superficial surgical site infections in the postoperative period. By processing a vast amount of data—389,865 surgical cases, 3,983,864 unlabeled chart notes, and 1,231,656 labeled notes—the stand-alone ML model achieved a sensitivity of 0.604, a specificity of 0.996, a positive predictive value (PPV) of 0.763, and a negative predictive value (NPV) of 0.991. When a human-in-the-loop pipeline was introduced, some values improved, such as sensitivity and NPV (increasing to 0.854 and 0.997, respectively), while specificity and PPV decreased (specificity dropped to 0.987, PPV to 0.603). Despite this, the human-in-the-loop approach was still more cost-effective and less time-consuming than the manual curation. Da Silva et al. [38] also used a similar ML system, using text mining from operative and postoperative reports to predict the risk of infections, and for their identification achieving good results with the Stochastic Gradient Descent model, achieving an ROC-AUC of 79.7% for prediction and the logistic regression model reaching an ROC-AUC of 80.6% for detection. Cho et al. [42] developed and tested several ML models to detect SSIs and discovered that integrating a rule-based algorithm with a ML algorithm and using 19 variables extracted from 1652 surgical cases, instead of the original 29, significantly improved SSI surveillance after colon surgery, as the combination reduced the need for manual chart reviews (−83.9%) while maintaining a high sensitivity.

Three separate articles [14,23,26] shared the common goal of preventing SSIs in patients undergoing some type of spinal surgery. Wang and collaborators [23] developed and validated a supervised Naïve Bayes algorithm that predicted the risk of infections in patients who underwent minimally invasive transforaminal lumbar interbody fusion, using only readily available data from 705 patients, achieving an AUC of 0.78. Hopkins et al. [14] developed a Deep Neural Network algorithm able to predict the risk of SSIs in patients undergoing spinal surgery, with a PPV of 92.56%. The algorithm also identified the top five risk factors—congestive heart failure, chronic pulmonary failure, hemiplegia/paraplegia, multilevel fusion, and cerebrovascular disease—and, surprisingly, even some protective factors. Finally, Liu and contributors [26] conducted a retrospective analysis of data from 288 patients who underwent spinal surgery and developed the most effective predictive algorithm, the XGBoost model, which achieved an AUC of 0.926.

In a study using both administrative and electronic medical records data from 27,360 surgical admissions (including 16,561 total knee arthroplasties and 10,799 total hip arthroplasties), Wu et al. [37] developed nine XGBoost machine learning models to automate SSI detection, distinguishing between superficial/deep incisional and organ space infections. The top model showed an impressive performance, with an ROC area under the curve (AUC) of 0.906 and a Precision–Recall (PR) AUC of 0.637, highlighting how effective machine learning algorithms can be in automating the detection of complex SSIs. Similarly, Flores-Balado et al. [46] worked on preventing post-hip replacement infections by creating a multivariable algorithm that used NLP and extreme gradient boosting to screen orthopedic patients and identify key SSI markers. The model, which was tested on data from 7444 surgeries, performed exceptionally well, achieving a sensitivity of 99.18%, a specificity of 91%, and a negative predictive value of 99.98%. When this system was integrated into the hospital’s routine, it notably reduced the time spent on surveillance (from 975 person-hours to 63.5 person-hours) and cut down the volume of manual reviews by 88.95%.

Focusing on both deep and superficial SSIs, Rafaqat et al. [43] aimed to develop a machine learning model capable of predicting both the type of SSI, as deep infections require more intensive treatment and higher costs, and the timing of when these infections would develop. The best model for predicting the type of infection was the extreme gradient boosting (XGBoost) univariate model, which achieved an AUC of 0.84 and a positive predictive value of 0.94. For predicting the week in which the SSI would develop, five out of twelve models reached the highest accuracy, each with an AUC of 0.74.

Additionally, other studies have worked on developing different AI models and ML algorithms to identify, and in some cases, predict and prevent, surgical site infections, mainly by analyzing data from electronic health records [18,20,22,34,35,36].

#### 3.2.2. Healthcare-Associated Pneumonia—HCAP

Epidemiological data show that healthcare-associated pneumonia (HCAP) is the most dangerous nosocomial infection, being the deadliest and the second most frequent type of HAI. HCAP is subdivided into hospital-acquired pneumonia (HAP) and ventilator-associated pneumonia (VAP), with the latter making up the majority of HCAP cases [48]. Prevention is therefore essential to avoid higher mortality rates and the increased costs associated with the impact on quality of life.

Kuo et al. [44] focused on this very aspect, developing a machine learning model to predict HAP in schizophrenic patients under anti-psychotic drugs, using 11 predictive factors. Of the seven models tested, the random forest model delivered the best results, with an AUC of 0.994. The C5.0 forest tree model also performed well, with an AUC of 0.993. The algorithm also identified the six major risk factors for developing the infection in these patients: medication dosage, clozapine use, duration of medication, changes in neutrophil and leukocyte counts, and drug–drug interactions.

Another important aspect of nosocomial infections is the presence of antibiotic-resistant microorganisms, which are particularly difficult to treat. Given the challenges clinicians face in selecting appropriate antibiotic therapy for methicillin-resistant Staphylococcus aureus (MRSA) and its potential to easily spread to other patients, Hirano et al. [40] developed a machine learning algorithm designed to predict MRSA infections in patients on mechanical ventilation. Using data extracted from the MIMIC-IV database, the XGBoost model was able to predict MRSA screening positivity with an AUROC of 0.89, a sensitivity of 0.98, and a positive predictive value of 0.65; the values regarding specificity were, however, low, only reaching 0.47.

Sophonsri et al. [19] also addressed the challenge of pneumonia in ventilated patients, focusing on identifying risk factors associated with the development and mortality of HCAPs. Their goal was to use this information to improve patient treatment and enhance antibiotic stewardship. The machine learning model they developed was based on data from 457 patients, subdivided by infection type (non-ventilated hospital-acquired pneumonia, ventilated hospital-acquired pneumonia, and ventilator-associated pneumonia). The model identified key risk factors for both the development of VAP (alcohol use disorder, APACHE II score at diagnosis, positive cultures for ESBL-Enterobacterales, and the need for vasopressor therapy before infection) and mortality (recent hospitalization within the last 30 days, active malignancy, isolation of ceftriaxone-resistant pathogens, or Pseudomonas aeruginosa and vasopressor therapy). Also, the AUC ROC values for mortality prediction in nvHABP, vHABP, and VABP models were 0.80, 0.78, and 0.83, respectively.

#### 3.2.3. Hospital-Acquired Urinary Tract Infections—HA-UTI

Nosocomial urinary tract infections are the most common type of HAIs, making up 40–60% of all cases, typically caused by the use of urinary catheters, either urethral or suprapubic. [48,49]. These infections are often complicated, sometimes leading to urosepsis, and diagnosing them correctly can be challenging [49].

Most of the articles reviewed focus on the early identification of risk factors for developing UTIs immediately upon hospital admission, with the ultimate goal of predicting an individual patient’s risk of acquiring a HA-UTI before it occurs [13,24,28,29,31]. Jakobsen et al. [29,31] conducted two studies focusing on the early detection of UTIs within 24 h of hospital admission. The 2023 study [29] used a Deep Neural Network model, achieving AUCs of 0.758 and 0.746 on full and reduced datasets, respectively, while the 2024 [31] study applied seven machine learning algorithms, particularly Bayesian Networks, with AUC values ranging from 0.720 to 0.746. In Denmark, Møller and collaborators [28] developed two predictive models for HA-UTIs within 48 h of admission using decision trees. They used both admission data and historical records from 301,932 patients, with the models achieving a good performance, with an ROC of 0.81 for the admission model and 0.74 for the 48 h model. Zachariah et al. [13] developed two predictive models, neural networks and decision trees, for assessing UTI risk at admission. While they achieved good sensitivity, specificity, and negative predictive values, their models had low positive predictive values, with 3.5% for the decision tree and 4.9% for the neural network. Finally, a study by Zhu et al. [24] in China developed a predictive system for UTIs in bedridden post-stroke patients, with the most effective model being an ensemble-learning model that achieved an AUROC of 82.2% in internal validation and 80.8% in external validation.

#### 3.2.4. Hospital-Acquired Infections—HAIs

Different AI and ML models have been developed and validated to make the detection of HAIs more automated and efficient, with the goal of reducing costs in terms of time, quality of life, and resources. Some approaches have focused on early identification at the time of hospital or ICU admission [17], including the risk assessment of multi-resistant bacterial colonization [32]. In other cases, the main focus was on patient monitoring, aiming to develop algorithms that could automatically and correctly identify the infection, improving the surveillance performance [39].

When a patient is infected with an HAI-causing agent, the high likelihood of these microorganisms being multi-drug resistant underscores the critical importance of proper antibiotic management. Bolton et al.’s main objective was the improvement of antimicrobial stewardship by developing a machine learning model to optimize and personalize the transition from intravenous (IV) antibiotics, which carry a higher risk of catheter-related infections, to oral antibiotics. Using data from approximately 10,000 hospital stays from two distinct datasets, 10 key features were identified to help determine the appropriate timing for this switch. The most effective model predicted when a patient could theoretically transition from IV to oral antibiotics, achieving an AUROC of 0.80 [45].

If an infection is not identified early, it may lead to bacteremia, which can then progress to sepsis; in such cases, timely detection of sepsis is crucial [50]. Lind et al. developed two machine learning models for the early detection of high-risk bacteremia leading to sepsis in vulnerable patients, with the full decision support tool achieving an AUC of 0.85 and the clinical factor-specific tool an AUC of 0.72 [16]. In the study conducted by Li et al., a random forest model identified the key negative prognostic factors associated with mortality in hospitalized patients suffering from an invasive candida infection alongside bacteremia, using data from 246 cases [25].

Intravascular devices, due to their direct access to the bloodstream and potential for prolonged use, are a major source of healthcare-associated bloodstream infections, which account for a significant portion of HAIs and carry a 10-to-20% mortality rate, along with an economic cost of approximately 40,000 USD per survivor [48]. Montella et al. utilized logistic regression and multi-layer perceptron models to predict which neonates were at risk of developing central line-associated bloodstream infection (CLABSI), while also identifying various risk factors leading to infection [33]. When a catheter-related infection is detected, clinicians face a critical decision: either save the catheter with antibiotic therapy or replace it. To support this decision-making process, Walker et al. developed four predictive models using machine learning algorithms to evaluate the individual risk of future CVC reinfections, helping a clinician choose between catheter salvage and replacement in pediatric patients [15]. A study from China [27] took a different approach by focusing on healthcare workers instead of patients. Huang et al. created an AI-powered system with a camera and speaker to monitor and improve the use of personal protective equipment and hand hygiene. Healthcare workers performed their tasks in front of the camera, and the AI reviewed the footage to check for proper practices, together with a human review. If any mistake was detected, a speaker alerted the staff and provided immediate feedback and training. After introducing this system, hospital infections dropped significantly from 1.31% in 2019 to 0.38% in 2022, and correct PPE use among 163 staff members improved from 52.12% to 98.14%.

## 4. Discussion

The analysis clearly highlights the potential of artificial intelligence and machine learning as valuable emerging tools for preventing and identifying healthcare-associated infections. HAIs pose a significant challenge to healthcare systems globally, impacting mortality and patient quality of life, and incurring substantial economic costs [2]. The integration of AI-based solutions can significantly mitigate these issues, as AI models can analyze extensive healthcare data using advanced algorithms to extract valuable clinical insights, while continuously improving their accuracy through adaptive learning and feedback [51].

Our review identified 35 articles where AI and ML were applied, with some focused on the prevention and others on the early identification of HAIs. In almost all cases, these methods demonstrated a strong performance in terms of AUROC, sensitivity, and specificity. A high sensitivity in detecting HAIs is preferred, especially for high-risk patients like those in the ICU or with compromised immune systems, to initiate prompt treatment and prevent complications such as sepsis. On the other hand, a high specificity is crucial for effective resource management, avoiding unnecessary treatments, and preventing antibiotic resistance. It ensures only true infection cases are treated and reported, maintaining accuracy in hospital metrics and avoiding inflated HAI rates. The AUROC score helps hospitals balance these competing priorities by evaluating the performance of AI-driven surveillance systems. A high AUROC value (closer to 1.0) reflects a system’s ability to accurately distinguish between infected and non-infected patients, allowing hospitals to fine-tune AI models for an optimal balance between sensitivity and specificity based on clinical and administrative needs.

We recorded an important heterogeneity between the articles themselves: most of the studies came from the United States and China, SSIs received much more attention than other infections, and, in some cases, the populations studied could be considered very “niche”.

Despite this heterogeneity, this review underscores that AI and ML models can significantly enhance the early and precise identification of high-risk patients, leading to more effective targeting of infection prevention measures in healthcare settings, ultimately reducing both incidence rates and associated costs. Firstly, the use of AI allowed for more efficient and accurate surveillance of infections, allowing infections to be identified early and preventive measures to be implemented quickly. Alternatively, AI could predict the risk of HAIs by identifying personalized risk factors for patients, thereby preventing the problem at its source. The adoption of automated surveillance systems has allowed AI to significantly reduce the workload of doctors and healthcare professionals by quickly processing large volumes of data and providing detailed analysis, compared to previous methods of manual monitoring [52]. Only a couple of studies reported an implementation of the developed models; however, when effectively implemented within the hospital department, AI and ML models effectively reduced manual workload [46], in one case by almost 85% [42], and reduced the incidence of HAI by 1.31% at 0.58% [27]. Another positive result achieved is the improvement of current early warning scores, such as SAPS II, improving the accuracy in predicting the mortality of these patients [32].

Thanks to their wide adoption, electronic health records (EHRs) are highly valuable due to the vast amount of data they store, which is essential for training AI and ML models, as demonstrated by the fact that almost every single study used data deriving from there. EHRs are far superior to administrative data, such as International Classification of Diseases (ICD) codes, which can often be unreliable and fail to consider clinical context [53]. A significant challenge, however, is the presence of unstructured data within EHRs such as clinical signs and symptoms. One effective approach to address this issue is using NLP ML models to analyze free text, which has proven particularly useful for identifying surgical site infections (SSIs) [12,30]. Another problem concerns the availability of EHRs, whose adoption is widespread especially in industrialized Western countries, while in developing countries their use is still limited mainly due to high costs and inadequate infrastructure [54].

AI can support monitoring and training on the correct use of PPE and hand washing, as highlighted in the study by Huang et al. [27] and in a review on the impact of intelligent environments and robots in preventing infections. However, the current instrumental limitations of AI and the poor compliance of healthcare workers, who may oppose daily tracking, still represent a challenge [55].

There are, however, numerous challenges and problems that need to be overcome in the coming years to fully harness the power of AI. Some challenges are purely technical in nature, including limitations of artificial intelligence models, and the need for a continuous flow of high-quality, complete, valid, and standardized data. The more that diagnostic/therapeutic algorithms differ from each other, the less practical data sharing between hospitals becomes, and the less valid the results obtained are.

The effectiveness of AI is constrained by the challenge of acquiring large, high-quality, and diverse datasets, as its performance is only as strong as the data used for training. Data completeness is also vital; for instance, automated surveillance in outpatient hemodialysis centers missed many bloodstream infections due to the absence of blood culture data in the dialysis EHRs. [53]. This is not to mention that accurate medical record documentation is essential for quality patient care, as the quality of medical records is closely linked to patient outcomes, while inaccuracies can compromise patient safety and increase the risk of malpractice [56].

Furthermore, the integration of health data in different settings is also an issue, as large-scale data sharing between healthcare facilities is essential but remains unachieved. Incomplete post-discharge surveillance can severely underreport HAIs. Linking EHRs across multiple inpatient and outpatient settings in the future could facilitate interfacility surveillance, increasing the detection of HAIs in non-rehospitalized patients or those readmitted elsewhere [53]. AI systems seem proficient at integrating diverse health data, such as clinical records, laboratory results, lifestyle information, and environmental factors; this enables healthcare providers to design more personalized and comprehensive treatment plans, improving the precision and effectiveness of care delivery [57]. However, biases in the data could skew AI models, leading to inaccurate treatment plans.

The size of the dataset is crucial: validating AI with small datasets can limit its ability to accurately differentiate between normal and abnormal variations and adequately address confounders, hindering the performance [58]. Unbalanced datasets (a high number of healthy people compared to infected ones) can pose an obstacle, as the data tend to be biased toward classifying the dominant class. In most cases analyzed, this problem was overcome by using a different machine learning algorithm or by applying techniques such as oversampling or undersampling [20].

Security and privacy represent significant challenges in AI implementation. AI systems are vulnerable to cyber-attacks, which can lead to misuse or fraud, making the safeguarding of these algorithms crucial as AI adoption grows. [58]. In the European Union, the General Data Protection Regulation (GDPR) places strict requirements on data ownership and consent, ensuring that patients control their own data. Explicit patient consent is required for the use and sharing of data, with full transparency about who has access, and how it will be stored, used, and protected [59].

The economic aspect of implementing AI also cannot be ignored. Although the costs in terms of financial expenditure are extremely high [4,5], to our knowledge no studies have analyzed the cost/benefit ratio of implementing AI in combating HAIs. Furthermore, none of the studies in this review indicated the costs for developing the AI models or how much the hospitals saved, nor any estimation on the cost of developing and applying AI in healthcare. It was predicted that the implementation of AI for diagnosis and treatment in hospitals would, over ten years, lead to savings of 15.17 h/day and 122.83 h/day. Translated in economic terms, savings would amount to 17,881 USD and 289,634 USD per day per hospital, respectively [60]. In other studies, estimated savings ranged from 200 billion USD to 360 billion USD in the United States [61]. However, it is important to consider the financial availability of smaller hospitals, which may struggle to cover the initial cost.

The question of determining liability in the use of AI is also particularly complicated, as clinical staff are traditionally accountable for their own decisions. The integration of AI into decision making introduces questions of accountability in cases of negative outcomes, potentially implicating clinicians, software developers, vendors, healthcare institutions, or regulators. The element of causation is highly case-specific; however, the inherent opacity of AI systems can pose significant challenges for patients trying to establish causation. This issue is further complicated by the difficulty of clearly explaining the algorithmic details to patients, which is often impractical or unfeasible [62]. Legal and ethical concerns surrounding these issues remain unresolved, with healthcare professionals currently held liable for decisions made with AI, even when they have limited understanding or control over the technology. Conversely, if a professional disregards AI recommendations and there is a resulting poor outcome, it could be viewed as clinical negligence [58]. The potential for serious complications arising from AI-driven decisions may discourage physicians, as they may be reluctant to take responsibility for outcomes associated with technologies they do not fully comprehend.

Poor availability or commitment of healthcare workers, due to limited time, low technological literacy, and, in some cases, reluctance to understand or use AI, could also lead to a slow or lack of adoption [58].

Definitions of healthcare-associated infections (HAIs) mainly focus on bacterial infections for epidemiological research, but other types of infections, particularly in the medico–legal context, require a clear causal link to healthcare settings. This distinction could impact the development of AI algorithms, as an infection could be classified as an HAI or not [63].

Finally, we highlight a potential problem in the development of an AI model in infection prevention in the form of a publication bias favoring “positive” results, as all analyzed studies show a decline in hospital-acquired infection rates. Consequently, suboptimal outcomes from AI models may go unpublished, leading to a distorted view of AI’s effectiveness in this area.

## 5. Conclusions

While AI cannot completely replace clinicians yet, the value of AI technology cannot be ignored. The integration of AI and ML into healthcare offers significant opportunities to improve the prevention and identification of healthcare-associated infections. The potential benefits regarding prevention, surveillance, diagnostic and therapeutic accuracy, reduction in the workload of healthcare personnel, and economic savings have been well described and, in some cases, tested and validated in the field. Despite this theoretical and practical potential, the application of AI in the field of hospital infections and healthcare in general still faces numerous barriers that hinder the adoption of these advanced technologies. The hope is that, in the years to come, many of these limitations will be overcome, enabling the implementation of a system that, with minimal expense, can enhance patient care and infection management.

## Figures and Tables

**Figure 1 healthcare-12-01996-f001:**
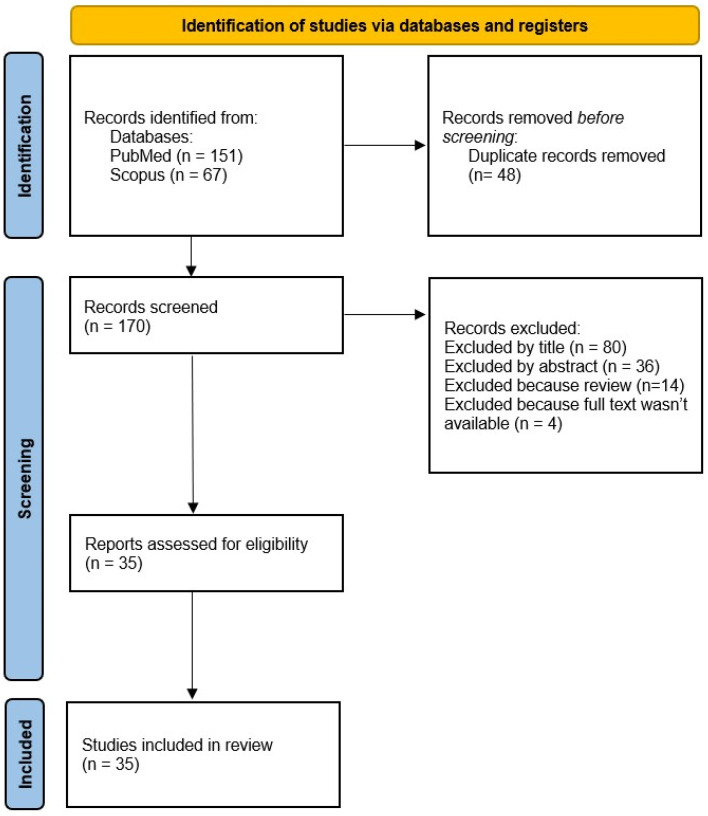
PRISMA flow diagram.

**Figure 2 healthcare-12-01996-f002:**
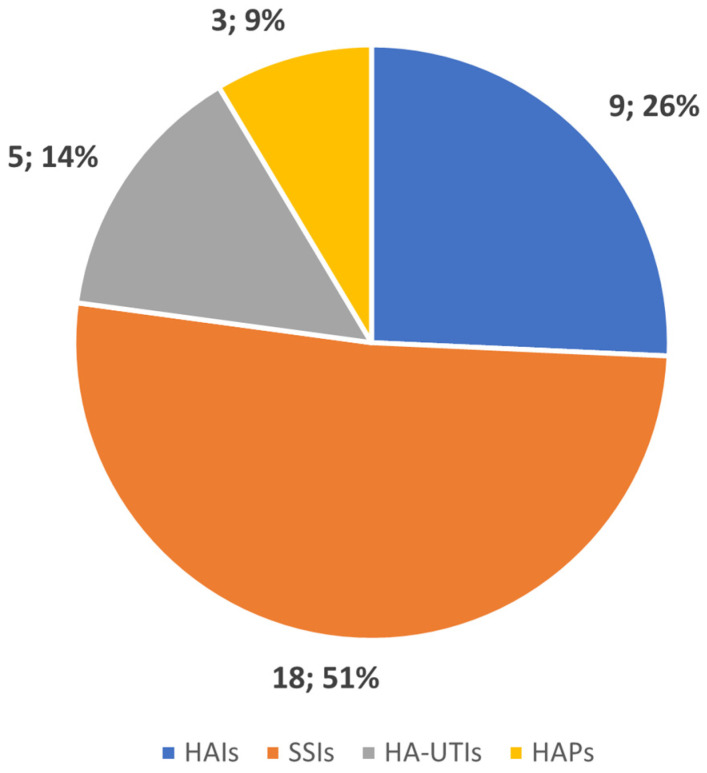
Number of studies conducted by type of HAI.

**Table 1 healthcare-12-01996-t001:** Characteristics of the studies included in the review.

Study	Year	Country	Study Setting	Studied Population	Infection Type	Aim of the Study	How the AI Was Trained	Results
Detection of Clinically Important Colorectal Surgical Site Infection using Bayesian Network [12]	2017	USA	General surgery	Patients that underwent colorectal surgery	Surgical site infection	Develop a Bayesian network automated detection system to identify surgical site infections after colorectal surgery	A Bayesian network was used to detect SSIs by utilizing risk factors from ACS-NSQIP data and keywords extracted from clinical notes through natural language processing(NLP) on data from 751 colorectal surgery cases. Two surgeons also provided the Bayesian network information on how to identify clinically meaningful SSIs (CM-SSIs)	The Bayesian network detected SSIs with an ROC of 0.827, which increased to 0.892 with surgeon-identified CM-SSI.
Novel Strategies for Predicting Healthcare-Associated Infections at Admission: Implications for Nursing Care [13]	2020	USA	Hospital	All patients	HA-UTI	Development of two machine learning models (neural networks and decision trees) to predict a patient’s risk of developing a UTI using data available on the first day of admission	Data from electronic health records of 897,344 hospitalized patients between 1 January 2009 and 31 December 2016	The decision tree model had a higher sensitivity compared to the neural network (78.2% vs. 57.3%), but it had a lower specificity (64.2% vs. 81.4%). The positive predictive values were 3.5% for the decision tree model and 4.9% for the Deep Neural Network model, while the negative predictive values were 99.4% and 99.1%, respectively
Using artificial intelligence (AI) to predict postoperative surgical site infection: A retrospective cohort of 4046 posterior spinal fusions [14]	2020	USA	Neurological and orthopedic surgery	Patients that underwent posterior spinal fusion surgery	Surgical site infection	Develop a ML model for the prediction of SSIs	Data from 4046 patients between 1 January 2000 and 31 December 2015	The Deep Neural Network model was able to predict postoperative SSIs with a PPV of 92.56% and an NPV of 98.45%, achieving a mean AUC of 0.775. It also helped identify risk factors and protective variables
Predicting outcomes in central venous catheter salvage in pediatric central line-associated bloodstream infection [15]	2021	USA	Pediatrics	Patients with central line-associated bloodstream infections (CLABSIs)	Healthcare-associated infections	Developing a machine learning approach to predict individual outcomes in central venous catheter (CVC) salvage	Data from electronic health records (patient demographics, diagnosis codes, medication records, infection history, laboratory data, all blood cultures, etc.) of 969 kids with CLABI between 2005 and 2018	The models predicting infection recurrence had higher AUROCs at most time points compared to the models for CVC removal (0.83 and 0.77 vs. 0.66 and 0.76)
Development and Validation of a Machine Learning Model to Estimate Bacterial Sepsis Among Immunocompromised Recipients of Stem Cell Transplant [16]	2021	USA	Hospital	Patients that were recipients of allogeneic hematopoietic cell transplant	Healthcare-associated infections	Development of two automated systems using ML that utilize electronic health records to predict potential sepsis in patients undergoing allogeneic hematopoietic cell transplant	Data from electronic health records of 1943 patients who underwent transplants between 2010 and 2019	The two systems (SHBSL and C-SHBSL) have sensitivities of 80% and 65.7%, respectively, and specificities of 72.8% and 66.9% in predicting the high risk of bacteremia for sepsis specifically in transplant patients
A Data-Driven Framework for Identifying Intensive Care Unit Admissions Colonized With Multidrug-Resistant Organisms [17]	2022	USA	ICU	All patients	HAI	Development of a data-driven framework to predict MRSA VRE and CRE colonization upon admission to ICU	Data from electronic healthcare records from University of Maryland‘s medical center of 3958 patients admitted to ICU from 2017 to 2018. in total 11 variables were included	The rate of colonization was 17,59% for MDRO, 13.03% for VRE, 1.45% for CRE, and 7.47% for MRSA. Sensitivity and specificity values with the best performing models, respectively were as follows: 80% and 66% for VRE with logistic regression, 73% and 77% for CRE with XGBoost, 76% and 59% for MRSA with random forest, and 82% and 83% for MDRO with random forest. It can be used as a clinical decision support tool for the identification of high-risk patients and the proper use of infection control measures
Developing an LSTM Model to Identify Surgical Site Infections using Electronic Healthcare Records [18]	2023	USA	Hospital	Patients that underwent surgery	Surgical site infection	Development of a long short-term memory (LSTM) model using electronic health record data to identify SSIs	Data from 9185 operative events from January 2016 to June 2021	The best model had an AUROC of 0.905
Machine learning to identify risk factors associated with the development of ventilated hospital-acquired pneumonia and mortality: implications for antibiotic therapy selection [19]	2023	USA	Hospital	Patients with pneumonia (non-ventilated, ventilated hospital-acquired, and ventilator-associated pneumonia)	Hospital-acquired pneumonia	Identify risk factors associated with the development and mortality of vHABP using ML	Data from inpatients who developed HAP (subdivided between vHABP, nvHABP, and VABP) between March 2014 and December 2019 (457 patients)	Identified the major risk factors for developing vHABP and the major risk for mortality. The AUC ROC of the nvHABP, vHABP, and VABP mortality models were 0.80, 0.78, and 0.83, respectively
Machine-learning models for predicting surgical site infections using patient pre-operative risk and surgical procedure factors [20]	2023	USA	Hospital	Patients that underwent surgery	Surgical site infection	Develop a ML model for the prediction of SSIs	Data from 2,882,526 patient records (sourced from the American College of Surgeons National Quality Improvement Program database) between 1 January 2013 and 31 December 2016	The Deep Neural Network model provided the best predictive performance with an area under the curve of 0.8518, accuracy of 0.8518, precision of 0.8517, sensitivity of 0.8527, and an F1-score of 0.8518
Predicting Surgical Site Infection after Colorectal Surgery Using Machine Learning [21]	2023	USA	Hospital	Patients who underwent colorectal surgery	Surgical site infection	Develop a ML model for the prediction of SSIs after colorectal surgery	Data from 275,152 patient records (sourced from the American College of Surgeons National Quality Improvement Program database) between 2012 and 2019	The Deep Neural Network model demonstrated the best performance for predicting SSIs, achieving an AUROC of 0.769 (95% CI 0.762–0.777). With a specificity of 50%, the sensitivity was 82%, while at a specificity of 90%, the sensitivity was 36%
Artificial Intelligence-Based Multimodal Risk Assessment Model for Surgical Site Infection (AMRAMS): Development and Validation Study [22]	2020	China	General surgery, gynecology, orthopedics, and urology	Inpatients that underwent surgery during the hospital stay	Surgical site infection	Development of an AI-based risk assessment model for surgical site infections (AMRAS)	Clinical data from electronic medical records (patient demographics, routine blood examination, coagulation, liver and kidney function, plasma electrolytes, smoking status, marital status, emergency intervention, and anesthesia) collected from patients who underwent a single operation between 2014 and 2019 (21,611 patients)	The AMRAS model identifies high-risk patients better than other machine learning methods and the currently used NNIS risk index
Development and Internal Validation of Supervised Machine Learning Algorithms for Predicting the Risk of Surgical Site Infection Following Minimally Invasive Transforaminal Lumbar Interbody Fusion [23]	2021	China	Hospital	Patients who underwent minimally invasive transforaminal lumbar interbody fusion	Surgical site infection	Develop and validate supervised ML algorithms for predicting the risk of SSI following minimally invasive transforaminal lumbar interbody fusion (MIS-TLIF)	Clinical characteristics, surgery-related parameters, and routine laboratory tests of 705 patients between May 2012 and October 2019	Naïve Bayes algorithm performed the best with an average AUC and ACC of 0.78 and 0.90
Prediction of post-stroke urinary tract infection risk in immobile patients using machine learning: an observational cohort study [24]	2022	China	Hospital	Immobile stroke patients	HA-UTI	Develop predictive models for UTI risk identification for immobile stroke patients.	The derivation cohort used data from 3982 immobile stroke patients between 1 November 2015 and 30 June 2016. The external validation cohort used data of 3837 patients collected from 1 November 2016 to 30 July 2017	The ensemble learning model had the best performance, with an AUROC during internal validation of 82.2% and of 80.8% during external validation and also had the highest sensitivity of 80.9% and 81.1% in both the internal and external validation sets
Machine-learning based prediction of prognostic risk factors in patients with invasive candidiasis infection and bacterial bloodstream infection: a singled centered retrospective study [25]	2022	China	ICU	Patients with invasive candida infection and bacterial bloodstream infection	HAI	Development of a machine learning algorithm for prognostic risk factors related to mortality in patients with candidiasis and bacteremia	Data from 246 hospitalized patients between 2013 and 2018	The ML identified the ten most important risk factors. The random forest model had the best AUC of 0.919
Using Preoperative and Intraoperative Factors to Predict the Risk of Surgical Site Infections After Lumbar Spinal Surgery: A Machine Learning Based Study [26]	2022	China	Hospital	Patients who underwent lumbar spinal surgery	Surgical site infection	Develop a ML model for the prediction of SSIs after lumbar spinal surgery	Data from 288 patients between December 2010 and December 2019	The XG Boost model had the best prediction performance with an average AUC of 0.926
Effectiveness of an artificial intelligence-based training and monitoring system in prevention of nosocomial infections: A pilot study of hospital-based data [27]	2023	China	Hospital	Hospital staff	HAI	Development of a camera/speaker system with integrated AI that monitors and provides training on the correct use of PPE and handwashing	Images taken from videos recorded directly by the system; subsequently, the “behaviors“ related to donning/removing PPE and handwashing were marked	After the introduction of the system, the accuracy of 163 operators increased from 52.15% to 98.14%. At the same time, the hospital infection rate decreased from 1.31% pre-AI to 0.58% in 2021 and 0.38% in 2022
Prediction of risk of acquiring urinary tract infection during hospital stay based on machine-learning: A retrospective cohort study [28]	2021	Denmark	Hospital	All patients	HA-UTI	Develop two predictive models utilizing data from the initial hospital admission and the patient’s historical records to predict the development of UTI upon hospital admission and within the following 48 h	Data collected from 301,932 patients between January 2017 and May 2018. Variables included demographic information, laboratory results, antibiotic treatment data, past medical history (ICD-10 codes), and clinical data	Both models (decision trees) had high ROC indices on the validation dataset: 0.81 for the Entry Model and 0.74 for the HAI Model, indicating adequate sensitivity and specificity. Both models could play a crucial role in personalized UTI prevention strategies for hospitalized patients
Clinical explainable machine learning models for early identification of patients at risk of hospital-acquired urinary tract infection [29]	2023	Denmark	Hospital	All patients	HA-UTI	Develop an ML model to predict patients at risk of HA-UTI using available data from electronic health records collected at the time of hospital admission	Data from electronic health records of 138,560 hospital admissions from 1 January 2017 to 31 December 2018	The Deep Neural Network model was the best-performing ML algorithm, with an AUC of 0.758 on a full dataset and an AUC of 0.746 on a reduced dataset. Within 24 h of admission, the ML model could identify patients at risk
Assessing the utility of deep neural networks in detecting superficial surgical site infections from free text electronic health record data [30]	2023	Denmark	Orthopedic surgery, general surgery, gynecology and obstetrics, urology, cardiothoracic surgery, ophthalmic surgery, plastic surgery, neurological surgery, otorhinolaryngology, oral and maxillofacial surgery, and vascular surgery	Patients who underwent surgery	Surgical site infection	Develop NLP on electronic health record chart notes to identify postoperative superficial surgical site infections	Deep Learning NLP models (stand-alone ML and a human-in-the-loop pipeline) were trained on data from 389,865 surgical cases between 1 January 2017 and 31 December 2021	The performance of the SAM pipeline was superior to administrative data (sensitivity of 0.604, specificity of 0.996, positive predictive value of 0.763, and a negative predictive value of 0.991). The HITL pipeline had a sensitivity of 0.854, a specificity of 0.987, a PPV of 0.603, and an NPV of 0.997
A study on the risk stratification for patients within 24 h of admission for risk of hospital-acquired urinary tract infection using Bayesian network models [31]	2024	Denmark	Hospital	All patients	HA-UTI	Using Bayesian Network-based machine learning models for risk stratification within 24 h of admission for HA-UTI to enable timely targeted preventive and therapeutic strategies	Information on 50 features (5 features in the reduced version selected based on expert-based knowledge) from 138,250 admissions from 1 January 2017 to 31 December 2018	The reduced clinical BN model (only five predictive features) reached the highest AUC of 0.746. The seven full BN models reached AUC scores between 0.746 and 0.720
A machine learning approach to predict healthcare-associated infections at intensive care unit admission: findings from the SPIN-UTI project [32]	2021	Italy	ICU	All patients	HAI	Development of a machine learning model that combines SAPS II with other patient data to predict the risk of ICU-associated infections at ICU admission	Data from the SPIN-UTI project (20,060 patients from 2006 to 2019). The algorithm combined SAPS II with other variables of the patients not already included in the SAPS II	AUC of 0.90 for the SAPS II + ML model
Predictive Analysis of Healthcare-Associated Blood Stream Infections in the Neonatal Intensive Care Unit Using Artificial Intelligence: A Single Center Study [33]	2022	Italy	Neonatal intensive care unit	Infants who stayed longer than two days in the level III NICU	HAI	Develop an AI model to predict whether a patient suffered from healthcare-associated bloodstream infection (HABSI)	Data (birth weight, gestational age, sex, length of stay, and duration of exposure to invasive devices) from 1203 neonates from 2016 to 2020	The logistic regression and multi-layer perceptron models achieved the highest AUC, accuracy, and F1-macro score in predicting HABSI (0.6027, 0.9461, and 0.6439, respectively). They also identified the most important risk factors in the development of HABSI
Risk Factors Analysis of Surgical Infection Using Artificial Intelligence: A Single Center Study [34]	2022	Italy	Hospital	All patients (excluding those with diabetes and those undergoing corticosteroid therapy)	Surgical site infection	Develop a logistic regression model to analyze the impact of various factors on the risk of SSIs. An AI model has then been employed to predict the risk of infection	Data from electronic health records (gender, age, length of stay, admission type, department, number of antibiotics, presence of surgical site infection) from 4031 patients who underwent surgery between 2015 and 2019	The K-Nearest-Neighbours model had the best performance, with a 94.9% accuracy and a 95.9% sensitivity and specificity
Predicting postoperative surgical site infection with administrative data: a random forests algorithm [35]	2021	Canada	Hospital	Patients that underwent surgery	Surgical site infection	Develop a ML model for the prediction of SSIs within 30 days after surgery	Administrative data from 14,351 patients who underwent surgery and enrolled in the National Surgical Quality Improvement Program from the Ottawa Hospital between 1 April 2010 and 31 March 2015	The full model demonstrated the best results with an AUC of 0.91 (95% CI, 0.90–0.92)
Validating administrative data to identify complex surgical site infections following cardiac implantable electronic device implantation: a comparison of traditional methods and machine learning [36]	2022	Canada	Hospital	Patients that underwent de novo cardiac implantable electronic device (CIED) implantation or generator replacement	Surgical site infection	Develop a ML model for the identification of SSIs using administrative data	Data from 3536 CIED procedures between 1 January 2013 and 31 December 2019	The ML model using administrative data achieved an AUC of 96.8%
Development of machine learning models for the detection of surgical site infections following total hip and knee arthroplasty: a multicenter cohort study [37]	2023	Canada	Hospital	Patients who underwent primary total elective hip (THA) or knee (TKA) arthroplasty	Surgical site infection	Develop a machine learning model to automate the process of SSI detection	Data from 22,059 patients (16,561 TKA and 10.799 THA) between 1 January 2013 and 31 August 2020	The optimal model achieved an ROC AUC of 0.906, PR AUC of 0.637, F1 score of 0.79, and sensitivity of 83.9%
Predicting the occurrence of surgical site infections using text mining and machine learning [38]	2019	Brazil	Hospital	Patients that underwent surgery	Surgical site infection	Develop a ML model that uses text mining to predict and detect SSIs	Data from 15,479 surgical descriptions and 12,637 postoperative records	The best performance for predicting SSI was achieved with the Stochastic Gradient Descent method, which had an ROC-AUC of 79.7%. For detection, logistic regression yielded the best performance, with an ROC-AUC of 80.6%
Automated healthcare-associated infection surveillance using an artificial intelligence algorithm [39]	2021	Brazil	Hospital	All patients	HAI	Develop an AI model for monitoring HAIs	Data spanning 18 months from the electronic health records of 5105 patients	The final model (multi-layer perceptron neural network), achieved an AUROC of 90.27%, with a specificity of 78.86% and a sensitivity of 88.57%. It accurately identified 67 out of 73 patients with HAIs and correctly classified 4637 patients as non-infected
Machine Learning Approach to Predict Positive Screening of Methicillin-Resistant Staphylococcus aureus During Mechanical Ventilation Using Synthetic Dataset From MIMIC-IV Database [40]	2021	Japan	Hospital	Mechanically ventilated patients	Hospital-acquired pneumonia	Development of a machine learning model that predicts MRSA infection in patients with mechanical ventilation	Data from the MIMIC-IV database (809 mechanically ventilated patients screened for MRSA)	The XGBoost model demonstrated superior performance in predicting MRSA screening positivity, achieving an AUROC of 0.89, sensitivity of 0.98, specificity of 0.47, and a positive predictive value of 0.65. Risk factors identified included admission through the emergency department (ED), central catheter placement, prior quinolone use, haemodialysis, and admission to the Surgical Intensive Care Unit (SICU)
Preliminary Evaluation of a Novel Artificial Intelligence-based Prediction Model for Surgical Site Infection in Colon Cancer [41]	2022	Japan	General surgery	Patients that underwent surgery for stage II–III colon cancer	Surgical site infection	Development of an AI model to predict the development of surgical site infections in patients with stage II–III colon cancer using immunological and nutritional markers	Data from 730 patients who underwent surgery for stage II–III colon cancer between 2000 and 2018	The accuracy of the AI model (area under the curve) is 0.73, comparable to previous risk prediction models using statistical analysis
Development of machine learning models for the surveillance of colon surgical site infections [42]	2024	South Korea	Hospital	Patients who underwent colorectal surgery	Surgical site infection	Develop ML models for the surveillance of SSIs for colon surgery	Data (two datasets with 26 and 33 variables) from 1652 surgical cases between January 2013 and December 2014	The Deep Neural Network model with Recursive Feature Elimination, utilizing 29 variables from the second database, achieved the highest performance, with an AUC of 0.963 and a PPV of 21.1%. By integrating a rule-based algorithm with a machine learning algorithm and reducing the variables to 19, the PPV increased to 28.9%. This hybrid method also reduced the number of cases needing manual review by 83.9% compared to the conventional method
Machine Learning Model for Assessment of Risk Factors and Postoperative Day for Superficial vs. Deep/Organ-Space Surgical Site Infections [43]	2023	Pakistan	Hospital	Patients that underwent surgery	Surgical site infection	Develop an ML model to predict type (superficial vs. deep) and timing of SSIs	Data from 113 patients from January 2019 to December 2020	The best model for predicting the type of SSI was the XGBoost univariate model, which achieved an AUC of 0.84, a PPV of 0.94, and an NPV of 1.57. For predicting the week of development of SSI, five models reached the same highest accuracy, with an AUC of 0.74
Predicting hospital-acquired pneumonia among schizophrenic patients: a machine learning approach [44]	2019	Taiwan	Mental hospital	Schizophrenic patients under anti-psychotic drugs	Hospital-acquired pneumonia	Develop a ML model for predicting hospital-acquired pneumonia among schizophrenic patients	Data (gender, age, clozapine use, drug–drug interaction, dosage, duration of medication, coughing, change in leukocyte count, change in neutrophil count, change in blood sugar level, and change in body weight) from medical records of 185 schizophrenic inpatients between 2013 and 2018	The random forest model gave the best results, with an AUC of 0.994, a sensitivity of 1000, and a specificity of 0.831. The study also identified the six most important risk factors for pneumonia among patients with schizophrenia
Personalising intravenous to oral antibiotic switch decision making through fair interpretable machine learning [45]	2024	UK	ICU	All patients	HAI	Develop a ML model to predict when a patient could switch from IV to oral antibiotics	Data from 10,362 unique ICU stays (8694 from the MIMIC dataset and 1668 from eICU); 10 clinical features were selected based on the UK antimicrobial IVOS criteria	The best model achieved a mean AUROC of 0.80 and could detect when an individual patient could switch from IV to oral antibiotics
Using artificial intelligence to reduce orthopedic surgical site infection surveillance workload: Algorithm design, validation, and implementation in 4 Spanish hospitals [46]	2023	Spain	Orthopedic surgery	Patients undergoing hip replacement surgery	Surgical site infection	Develop an AI model to predict SSIs in patients undergoing hip replacement surgery	Data from electronic health records of 6741 patients (7444 surgeries) between January 2014 and May 2024	The AI model (natural language processing and extreme gradient boosting) demonstrated a sensitivity of 99.18% and a specificity of 91%, with a negative predictive value of 99.98% and an AUC of 0.989. It has been integrated as a screening tool for postoperative patients, significantly reducing the volume of clinical records that need manual evaluation

## Data Availability

Data are contained within the article.

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
