# Peer review of "Advancing Patient Safety: The Future of Artificial Intelligence in Mitigating Healthcare-Associated Infections: A Systematic Review"

_healthcare, 2024, doi:10.3390/healthcare12191996_

Round 1

Reviewer 1 Report

Comments and Suggestions for Authors

The manuscript reviews the literature on the broad application of AI for healthcare-associated infection surveillance, prevention and control. The article is within the journal scope, original and interesting, so worthy of publication.

I have a major and some minor issues to be addressed, then I suggest "major revisions"

Major issue: the literature review was conducted only on Pubmed, while it is recommended to employ at least two datasets. I suggest integrating with another database.

Minor issues:

Title: maybe too broad compared to the aim AI application in HAI surveillance. Litigation topic is not discussed so please remove.

abstract: the conclusions are too broad compared to the results, please revise accordingly.

Introduction: I also suggest to mention the Point prevalence survey of healthcare-associated infections and antimicrobial use in European acute care hospitals – 2022-2023 which addresses (for Europe) for the first time the central issue of automation for surveillance of HAIs.

PRISMA checklist ok

line 116 Characteristics of Eligible Studies: please revise and integrate with the number of authors assessing eligibility. 

line 121 Quality Assessment and Risk of Bias: please revise and integrate as related to quality assessment of the included articles (for instance: all articles included after screening and revision were considered of high quality and complete) and risk of search biases (or add a limitation after considering the major issue aforementioned).

line 189 please revise the typo with references

from line 359: nosocomial infections: please verify this definition or if is better HAI. 

line 368 very relevant. Please discuss briefly the relevance of sensitivity and specificity regarding the purpose and perspective of analysis of the data.  Indeed, it might be preferable to have an excess of specificity or sensitivity according to a need for treatment (empirical therapy) by a clinician or instead for reporting at various levels by a medical director/health manager.

Among the critical factors starting at line 405, I would suggest, if deemed appropriate, to discuss further the issue of the feeding of the datasets with which AI and ML are developed. In fact, in addition to the issue of digitisation in the premise (and the spread of EHRs), there remains the question of data quality and completeness (as partially addressed previously by the authors in 10. 2174/1389201020666190408102221), the integration between systems to extract with certainty the data for infection definition and infection prevention and control policies (laboratory, clinic, diagnostic, therapy, safety checklists, administrative data, etc.); the integration of health data in different settings (from hospitals, family medicine, nursing homes and home monitoring and care); the safety and compliance risks and the purpose of collecting and using the sensitive data, in relation to patient authorisations;  the bias of publishing only the "positive" data (if the infection trend is increasing, no one will publish); the prevalence of study on bacteria and recently on Sars-Cov-2. This has implications for the algorithm used to define an infection as an HAI (see 10.1007/s40506-020-00216-7 and even if limited 10.7416/ai.2024.2603. Epub 2024 Jan 18).

line 420 between parentheses, redundant, please remove as yet stated in the introduction.

line 431 The wording is too colloquial. The authors point out that data from the literature are scarce and mostly support and simplify the decision maker (clinician or manager). Hence, the question of overall responsibility for AI, considering the different purpose and results of the review but the medico-legal authors, could be discussed a bit. Also interesting is the documentation of AI outcomes in the clinical record, such as the automated preoperative risk assessment for infections.

Many thanks.

Author Response

Dear Reviewer 1,

Thank you very much for dealing with our manuscript titled “Advancing Patient Safety: The Future of Artificial Intelligence in Mitigating Healthcare-Associated Infections (HAI) and Medico-Legal Litigation through Automated Prevention and Surveillance. A Systematic Review.”, and giving us the opportunity to revise it according to the criticism of the Reviewers. We would like to thank the reviewers for raising interesting points and constructive criticism, which we have all been addressed in our revised manuscript now entitled “Advancing Patient Safety: The Future of Artificial Intelligence in Mitigating Healthcare-Associated Infections: a Systematic Review”. As a result of this, the revised manuscript has been profoundly modified in the text, and we believe that it has improved considerably.

We below responded to the specific comments of the reviewers point by point and detailed all the modifications to the text.

We hope that with these revisions our manuscript has now reached a maturity that justifies publication

Sincerely yours,

Stefano D’Errico

Point-by-point response to Reviewer 1:

Comment: The manuscript reviews the literature on the broad application of AI for healthcare-associated infection surveillance, prevention and control. The article is within the journal scope, original and interesting, so worthy of publication.

Reply: We thank the Reviewer for the positive comments and the points raised that allowed us to profoundly change the manuscript.

Comment: The literature review was conducted only on Pubmed, while it is recommended to employ at least two datasets. I suggest integrating with another database. 

Reply: The search has been integrated with another database (Scopus database).

Comment: Title: maybe too broad compared to the aim AI application in HAI surveillance. Litigation topic is not discussed so please remove.

Reply: The title has been changed and shortened.

Comment: Abstract: the conclusions are too broad compared to the results, please revise accordingly.

Reply: The abstract has been updated, with particular attention given to the conclusions.

Comment: Introduction: I also suggest to mention the Point prevalence survey of healthcare-associated infections and antimicrobial use in European acute care hospitals – 2022-2023 which addresses (for Europe) for the first time the central issue of automation for surveillance of HAIs.

Reply: We thank the Reviewer for advising this article, which has been included in the Introduction (see lines 77 to 91)

Comment: Line 116 Characteristics of Eligible Studies: please revise and integrate with the number of authors assessing eligibility.

Reply: The subsection has been modified according to Reviewer’s comments (see lines 161-162).

Comment: Line 121 Quality Assessment and Risk of Bias: please revise and integrate as related to quality assessment of the included articles (for instance: all articles included after screening and revision were considered of high quality and complete) and risk of search biases (or add a limitation after considering the major issue aforementioned). 

Reply: The subsection has been modified according to Reviewer’s comments (see lines 165-168).

Comment: Line 189 please revise the typo with references

Reply: The typo has been corrected.

Comment: From line 359: nosocomial infections: please verify this definition or if is better HAI.

Reply: The term “nosocomial infections” has been changed to HAI, as is considered more inclusive and widely used in current medical discussions.

Comment: Line 368 very relevant. Please discuss briefly the relevance of sensitivity and specificity regarding the purpose and perspective of analysis of the data. Indeed, it might be preferable to have an excess of specificity or sensitivity according to a need for treatment (empirical therapy) by a clinician or instead for reporting at various levels by a medical director/health manager.

Reply: The relevance of sensitivity, specificity and also the AUROC has been integrated into the Discussion (see lines 409-422).

Comment: Among the critical factors starting at line 405, I would suggest, if deemed appropriate, to discuss further the issue of the feeding of the datasets with which AI and ML are developed. In fact, in addition to the issue of digitisation in the premise (and the spread of EHRs), there remains the question of data quality and completeness (as partially addressed previously by the authors in 10. 2174/1389201020666190408102221), the integration between systems to extract with certainty the data for infection definition and infection prevention and control policies (laboratory, clinic, diagnostic, therapy, safety checklists, administrative data, etc.); the integration of health data in different settings (from hospitals, family medicine, nursing homes and home monitoring and care); the safety and compliance risks and the purpose of collecting and using the sensitive data, in relation to patient authorisations; the bias of publishing only the "positive" data (if the infection trend is increasing, no one will publish); the prevalence of study on bacteria and recently on Sars-Cov-2. This has implications for the algorithm used to define an infection as an HAI (see 10.1007/s40506-020-00216-7 and even if limited 10.7416/ai.2024.2603. Epub 2024 Jan 18).

Reply: The various issues raised by the Reviewer have been addressed throughout the discussion, beginning from line 466. We also appreciate the Reviewer’s recommendation of articles, which have been incorporated into the discussion.

Comment: Line 420 between parentheses, redundant, please remove as yet stated in the introduction.

Reply: The text between parentheses has been removed.

Comment: Line 431 The wording is too colloquial. The authors point out that data from the literature are scarce and mostly support and simplify the decision maker (clinician or manager). Hence, the question of overall responsibility for AI, considering the different purpose and results of the review but the medico-legal authors, could be discussed a bit. Also interesting is the documentation of AI outcomes in the clinical record, such as the automated preoperative risk assessment for infections.

Reply: The wording was changed and the medico-legal aspects of the implementation of AI in healthcare were addressed more in depth (from line 509).

Reviewer 2 Report

Comments and Suggestions for Authors
In the proposed study: Advancing Patient Safety: The Future of Artificial Intelligence in Mitigating Healthcare-Associated Infections (HAI) and Medico-Legal Litigation through Automated Prevention and Surveillance. A Systematic Review, A review has been conducted of healthcare-related issues using AI. AI techniques in healthcare that have proven useful include the detection of clinical conditions in medical imaging and diagnostic services, however, so points are noted for the improvement of the quality of the paper.

1. The author needs to improve the figures’ quality.

2. There are many typos and grammatical mistakes in the writing. Also, the article must be checked for linguistic and grammatical errors. Please clear all the issues before resubmitting.

3. the analysis and results need to highlight the accuracy and efficiency of the proposed model. This approach shows significant potential in effectively managing the estimation process in systems with dynamic topological changes.

4. The literature missing important relevant research that needs to be included, i.e.

a. Arjmandnia, F., & Alimohammadi, E. (2024). The value of machine learning technology and artificial intelligence to enhance patient safety in spine surgery: a review. Patient Safety in Surgery18(1), 11.

b. Islam, U., Mehmood, G., Al-Atawi, A. A., Khan, F., Alwageed, H. S., & Cascone, L. (2024). NeuroHealth guardian: A novel hybrid approach for precision brain stroke prediction and healthcare analytics. Journal of Neuroscience Methods409, 110210.

5. The title of the paper is too long, you need to revise the title and remove "." from the ending
6.  The abstract is a little thin and does not quit convey the vibrancy of the findings and the depth of the main conclusions. The authors should please extend this somewhat for a better first impression.

7. All the references must be uniform and according to the journal format.

Comments on the Quality of English Language

Minor editing is needed.

Author Response

Dear Reviewer 2,

Thank you very much for dealing with our manuscript titled “Advancing Patient Safety: The Future of Artificial Intelligence in Mitigating Healthcare-Associated Infections (HAI) and Medico-Legal Litigation through Automated Prevention and Surveillance. A Systematic Review.”, and giving us the opportunity to revise it according to the criticism of the Reviewers. We would like to thank the reviewers for raising interesting points and constructive criticism, which we have all been addressed in our revised manuscript now entitled “Advancing Patient Safety: The Future of Artificial Intelligence in Mitigating Healthcare-Associated Infections: a Systematic Review”. As a result of this, the revised manuscript has been profoundly modified in the text, and we believe that it has improved considerably.

We below responded to the specific comments of the reviewers point by point and detailed all the modifications to the text.

We hope that with these revisions our manuscript has now reached a maturity that justifies publication

Sincerely yours,

Stefano D’Errico

Point-by-point response to reviewer:

Comment: The author needs to improve the figures’ quality.

Reply: The figures’ quality has been improved.

Comment: There are many typos and grammatical mistakes in the writing. Also, the article must be checked for linguistic and grammatical errors. Please clear all the issues before resubmitting.

Reply: The paper has been thoroughly checked for typos and grammatical errors, which have been corrected.

Comment: the analysis and results need to highlight the accuracy and efficiency of the proposed model. This approach shows significant potential in effectively managing the estimation process in systems with dynamic topological changes.

Reply: The authors conducted a literature review and do not intend to propose any AI model or intend to favor any model over others. The accuracy and efficiency of the various articles, and consequently the various AI models, are addressed in the Results section. The Discussion section (from line 409) provides an explanation of the importance of sensitivity, specificity, and AUROC values as presented in the previous section and in Table 1.

Comment: The literature missing important relevant research that needs to be included, i.e.

Arjmandnia, F., & Alimohammadi, E. (2024). The value of machine learning technology and artificial intelligence to enhance patient safety in spine surgery: a review. Patient Safety in Surgery, 18(1), 11.

Islam, U., Mehmood, G., Al-Atawi, A. A., Khan, F., Alwageed, H. S., & Cascone, L. (2024). NeuroHealth guardian: A novel hybrid approach for precision brain stroke prediction and healthcare analytics. Journal of Neuroscience Methods, 409, 110210.

Reply: We thank the Reviewer for advising these articles, which have been included in the Introduction (see lines 98-103)

Comment: The title of the paper is too long, you need to revise the title and remove "." from the ending

Reply: The title has been changed and shortened.

Comment: The abstract is a little thin and does not quit convey the vibrancy of the findings and the depth of the main conclusions. The authors should please extend this somewhat for a better first impression.

Reply: The abstract has been updated and lengthened, with particular attention given to the conclusions.

Comment: All the references must be uniform and according to the journal format.

Reply: The references have been made uniform and in accordance with the journal format.

Round 2

Reviewer 1 Report

Comments and Suggestions for Authors

The systematic review is robust and comprehensive. I have no further considerations. Sincerely

Author Response

Dear Reviewer,

thanks a lot. Your suggestions contributed to improve the manuscript.

Warm regards.

Stefano D'Errico

Reviewer 2 Report

Comments and Suggestions for Authors

The author sufficiently addressed my concerns, however, the reference lists need to include some of the latest articles related to the topic domain; such as,

1.  Dai, M., Sun, G., Yu, H., & Niyato, D. (2024). Maximize the Long-Term Average Revenue of Network Slice Provider via Admission Control Among Heterogeneous Slices. IEEE/ACM Transactions on Networking, 32(1), 745-760. doi: 10.1109/TNET.2023.3297883.

2. Zhu, C. (2023). Intelligent robot path planning and navigation based on reinforcement learning and adaptive control. Journal of Logistics, Informatics and Service Science, 10(3), 235-248. doi: 10.33168/JLISS.2023.0318.

Comments on the Quality of English Language

Just check for minor mistakes

Author Response

Dear reviewer,

thanks for your valuable suggestions.

Authors think that the two references are quite far from the aim of the manuscript, no more adding the content of the research.

I hope you can understand if we don't update the reference list.

We also check the text with mother tongue.

Best regards

Prof. Stefano D'Errico